# Arthritogenic Alphavirus-Induced Immunopathology and Targeting Host Inflammation as A Therapeutic Strategy for Alphaviral Disease

**DOI:** 10.3390/v11030290

**Published:** 2019-03-22

**Authors:** Helen Mostafavi, Eranga Abeyratne, Ali Zaid, Adam Taylor

**Affiliations:** Emerging Viruses and Inflammation Research Group, Institute for Glycomics, Griffith University, Gold Coast, QLD 4222, Australia; helen.mostafavi@griffithuni.edu.au (H.M.); e.abeyratne@griffith.edu.au (E.A.); a.zaid@griffith.edu.au (A.Z.)

**Keywords:** alphavirus, arthralgia, chikungunya, Ross River virus, emerging virus, mosquito-borne

## Abstract

Arthritogenic alphaviruses are a group of medically important arboviruses that cause inflammatory musculoskeletal disease in humans with debilitating symptoms, such as arthralgia, arthritis, and myalgia. The arthritogenic, or Old World, alphaviruses are capable of causing explosive outbreaks, with some viruses of major global concern. At present, there are no specific therapeutics or commercially available vaccines available to prevent alphaviral disease. Infected patients are typically treated with analgesics and non-steroidal anti-inflammatory drugs to provide often inadequate symptomatic relief. Studies to determine the mechanisms of arthritogenic alphaviral disease have highlighted the role of the host immune system in disease pathogenesis. This review discusses the current knowledge of the innate immune response to acute alphavirus infection and alphavirus-induced immunopathology. Therapeutic strategies to treat arthritogenic alphavirus disease by targeting the host immune response are also examined.

## 1. Introduction

Arthritogenic alphaviruses are a group of positive sense, single-stranded RNA viruses belonging to the genus *Alphavirus* of the *Togaviridae* Family. Alphaviruses are transmitted between non-human vertebrate reservoirs and humans by mosquito vectors. Members of the *Alphavirus* genus can be classified into two groups based on their historical geographic distribution and disease pathologies resulting from infection [1]. Encephalitic alphaviruses, commonly known as “New World” alphaviruses, such as Venezuelan equine encephalitis virus (VEEV), western equine encephalitis virus (WEEV) and eastern equine encephalitis virus (EEEV), predominantly cause life-threatening neuro-pathological manifestations in humans. Arthritogenic alphaviruses are commonly referred to as “Old World” alphaviruses and include viruses, such as chikungunya virus (CHIKV), o’nyong-nyong virus (ONNV), Ross River virus (RRV), Barmah Forest virus (BFV), Mayaro virus (MAYV), Sindbis virus (SINV) and Semliki Forest virus (SFV). Symptomatic infection with arthritogenic alphaviruses typically results in crippling arthritic joint manifestations.

Arthritogenic alphaviruses are globally distributed. RRV is endemic to Australia and the Pacific, ONNV is found in Africa, BFV is endemic to Australia and MAYV to South and Central America. CHIKV has emerged as a major global human pathogen over the last 10–15 years, with outbreaks in Southeast Asia (Thailand, Indonesia, Malaysia, Cambodia and Singapore), the Indian Ocean islands of Mauritius, La Réunion and the Seychelles (270,000 cases in La Réunion 2005–2006) and India (3–4 million estimated cases 2005–2011) [2,3,4]. CHIKV recently became established in the Americas, infecting over 1.5 million people in just 2 years [5]. This epidemic was aided by the emergence of a CHIKV clade transmitted by the mosquito *Aedes albopictus*, which has dramatically expanded the geographic distribution of CHIKV [6]. Viral vector adaptation, climate change, increased international travel and urban development are thought to be factors capable of expanding the range of alphaviruses and increasing their epidemic potential [7].

Patients with acute arthritogenic alphavirus infection typically present with high viraemia, fever, skin rash, myalgia and incapacitating arthralgia. Joint arthralgia and inflammation, being the predominant sign of acute disease, can range from tenderness to limited joint movement with extreme redness and swelling. Small joints of the hand, wrist and ankles and larger joints, such as the knee and shoulder, are frequently afflicted. The pain, likened to that experienced by patients with rheumatoid arthritis (RA), immobilises patients and greatly reduces the quality of life [8]. Although usually self-limiting, chronic or recurrent rheumatic manifestations have been reported following resolution of acute disease [9]. Cases of CHIKV infection have also reported neurological complications [10]. In this review, we have primarily focused on arthritogenic alphavirus-induced immunopathology and prospective therapeutic strategies to treat arthritogenic alphaviral disease. Much of the current understanding of alphavirus pathogenesis has been acquired from studies using established animal models of arthritogenic alphavirus infection.

## 2. Innate Immune Response to Acute Arthritogenic Alphavirus Infection

Arthritogenic alphavirus infections can cause severely debilitating and often chronic rheumatic disease. Although not typically life-threatening, infection with some alphaviruses, such as CHIKV, has also been associated with severe neurological disease, particularly, in the young and the elderly [11]. Thus, there is a pressing need to improve understanding of the immune mechanisms that control alphavirus infection. Interestingly, both innate and adaptive immune responses have been shown to play a major role in arthritogenic alphaviral disease, having both protective and pathologic mechanisms. Many invading pathogens are first recognised as foreign by the innate immune response. This fast acting, multifaceted system rapidly detects alphavirus infection and mobilises molecular and cellular responses that facilitate protective immunity. The innate immune response stimulates adaptive immune responses, which act to clear the infection and generate lasting immunological memory. The innate immune response to arthritogenic alphavirus infection is chiefly regulated by type I interferon (IFN). Induction of type I IFN (IFN-α/β) is a hallmark of the innate immune response to viral infection [12,13]. Specifically, IFN-α levels were significantly elevated in plasma of acute CHIKV-infected patients [14]. Alphaviruses efficiently induce type I IFN in small animal models [15]. Where wild type (WT) adult mice are resistant to CHIKV infection, mice with a partially (IFN-α/β receptor^+/−^) or totally (IFN-α/β receptor^−/−^) abrogated type I IFN pathway develop a mild or severe infection, respectively. Disease severity correlates with the degree of viral replication and tissue tropism, with virus observed in the joints and muscle of mice with partially abrogated type I IFN, and greater dissemination, including the central nervous system, in mice with a totally abrogated type I IFN pathway [16]. Type I IFN provides resistance to alphavirus infection and mice with defects in type I IFN signalling rapidly succumb to arthritogenic alphavirus infection [17,18,19,20,21,22].

The complement system and additional immune effectors also play a role in the initial response to alphavirus infection. An essential component of the innate immune response, complement, is involved in pathogen recognition, immunoregulation and has been demonstrated to have both an antiviral and pathogenic role during alphavirus infection. Complement has been shown to limit viraemia and reduce CNS titre of SINV in infected mice while shortening mouse survival time post-infection [23]. Morbidity was prolonged in complement-deficient mice infected with SINV. Mice deficient in C3, the central component of the complement system on which the classical, mannose-binding lectin (MBL) and alternative activation pathways all converge, developed much less severe inflammatory disease than WT mice infected with RRV [24,25]. CR3 deficiency had no effect on RRV replication but did reduce the expression of proinflammatory mediator, interleukin-6 (IL-6) [25]. Further studies have determined the MBL pathway as essential for the development of RRV-induced disease and that N-linked glycans of the E2 protein act as a primary ligand for MBL deposition and complement activation on RRV-infected cells [26,27].

Production of reactive oxygen species (ROS) during acute alphavirus infection stimulates a unique pathogen recognition receptor (PRR) that induces a host defence response to SINV [28]. The ROS-dependent oxidisation of the nuclear pore induces the redistribution of tetrachlorodibenzo-p-dioxin (TCDD)-inducible poly (ADP ribose) polymerase (TIPARP) into the cytoplasm where SINV replicates. TIPARP binds to SINV RNA via its zinc finger domain and recruits an exosome to induce viral RNA degradation [28]. Induced by both type I and type II IFNs, interferon regulatory factor I (IRF-1) is a key transcription factor in the host defence response against arthritogenic alphavirus infection [29]. In vivo experiments demonstrated that IRF-1 expression limited CHIKV-induced foot swelling in joint-associated tissues and prevented dissemination of CHIKV and RRV. IRF-1 also restricted CHIKV and RRV infection in stromal cells [29]. Inflammasomes, which include NOD-like receptor (NLR) family molecular complexes, are activated by alphaviral pathogen-associated molecular patterns (PAMPs). Inhibition of the inflammasome via caspase-1 silencing was found to enhance CHIKV replication in dermal fibroblasts [30]. NLRP3 activation by alphavirus infection led to potent IL-1β and IL-18 production [31]. Treatment of mice with NLRP3 or caspase-1 inhibitors led to less severe disease in CHIKV and RRV infected mice [31].

### 2.1. Induction of Interferon and Alphavirus Detection

Alphavirus replication in the host cell leads to elevated production of PAMPs, which ultimately induce the expression of type I IFN. During replication of the single-stranded RNA genome, partially double-stranded (ds) RNA replicative intermediate PAMPs accumulate and are recognised as foreign by host PRRs. PRRs include RNA-sensing toll-like receptors (TLRs) and members of the cytosolic retinoic acid-inducible gene I (RIG-I)-like receptors, such as melanoma differentiation-associated protein 5 (MDA5) and RIG-I, that detect alphaviruses through their genomic RNA or dsRNA intermediates [32]. Typically, PRRs activate a signalling cascade resulting in the expression of transcription factors or IRFs responsible for the induction of IFN and an antiviral state.

Mice deficient in endosomal TLR signalling (TLR3, TLR7 and TLR9) showed heightened susceptibility to infection with a neuroadapted SINV [33]. Despite this, survival from infection was largely independent of these TLR-driven responses, with redundancy in the pathways driving type-I IFN production. In vivo analysis of innate immune mechanisms of the CNS demonstrated prevalent TLR expression with TLR3 found in abundance in the resting CNS. Upon SFV and CHIKV infection, TLR gene expression was significantly up-regulated, correlating with IFN-α production [34,35]. TLR3 and TLR9 up-regulation was critically dependent on the presence of a functional IFN-α/β system [34]. SFV infected glial cells, particularly astrocytes, showed strong up-regulation of several TLR genes in vitro [36]. Again, TLR3 showed the largest increase with TLR8 and 9 also highly up-regulated [36].

TLR3 has been shown to regulate host immunity to CHIKV. Mice deficient in TLR3 showed increased virus titres and dissemination with aggravated pathology marked by increased proinflammatory myeloid cell infiltration [32]. Interestingly, TLR3 was found to modulate the early immunoglobulin (Ig)G response to CHIKV infection with antibodies from TLR3^−/−^ mice exhibiting low in vitro neutralisation capacity [32]. Results demonstrate that a TLR3-mediated antibody response to CHIKV infection is required to control virus replication and disease. Increased induction of TLR3 and IFN-β in mice, pretreated with the TLR-3 agonist and IFN inducer poly I:C (Polyinosinic:Polycytidylic acid), reduced CHIKV titre in the mouse brain [35]. This protection was associated with the increased expression of antiviral genes. Studies on the TLR3 downstream signalling protein Toll/IL-1R domain containing adaptor inducing IFN-β (TRIF) provide further evidence of the role of TLR3 in IFN induction and protection from alphavirus infection. SINV-infected TRIF^−/−^ mice showed increased susceptibility to infection compared to WT infected mice [37]. Interferon-activating small molecules that stimulate the TRIF-dependent signalling cascade strongly inhibited CHIKV replication [38]. TRIF^−/−^ mice infected with CHIKV had increased viraemia and foot swelling and was significantly more pronounced than in WT infected mice [39]. The same study investigated the role of the TLR7 signalling protein myeloid differentiation primary response gene 88 (MyD88). CHIKV-infected MyD88^−/−^ mice also developed high viraemia; however, no difference in disease profile compared to WT infected mice suggests the MyD88 pathway has a less important role in the antiviral response to CHIKV [39]. Similar findings were observed between TRIF^−/−^ and MyD88^−/−^ mice during SINV infection [37]. Other studies have shown the role of TLR3 in CHIKV control to be negligible in contrast to MyD88 [22]. Myd88-deficient mice infected with CHIKV, like TLR3^−/−^ mice, did not succumb to infection but developed significant viral dissemination [22]. Myd88-dependent TLR7 signalling was significantly involved in protection from severe RRV-induced arthritic disease [40]. RRV-infected mice deficient in Myd88 or TLR7 developed more severe disease and mortality than WT infected mice [40]. Both Myd88^−/−^ and TLR7^−/−^ mice exhibited higher viral loads than WT mice at late times post-infection. Furthermore, RRV-specific antibody produced in TLR7-deficient mice had little neutralising activity. Myd88^−/−^ and TLR7^−/−^ mice also showed defects in germinal centre activity, suggesting TLR7-dependent signalling is critical for the development of protective antibody responses against RRV [40]. Interestingly, Myd88 was found to have no role in the protection or control of neurovirulent SINV pathogenesis [33,37].

RIG-I-like helicase (RLH) family members, RIG-I and MDA5, play important roles in inducing the type I IFN pathway during alphavirus infection. PRRs, RIG-I and MDA5 proteins detect cytosolic dsRNA or 5’-triphosphate RNA ligands and mediate IFN production. SINV infection and expression of the SFV replicase alone, which converts host cell RNA to 5’-triphosphate and dsRNA, were able to induce type I IFN through the RIG-I and MDA5 pathways [41,42]. RIG-I has been shown to associate specifically to the 3’ untranslated region of the CHIKV genome during infection [43]. RIG-I is up-regulated during RRV infection, and stimulation of the RIG-I pathway with 5’-triphosphate RNA generated a robust antiviral response against CHIKV that was reliant on the transcription factor IRF3 [44,45]. Small molecule activators of the IRF3-activating adaptor molecule stimulator of IFN genes (STING) inhibited CHIKV and SINV replication and blocked CHIKV viraemia [46]. The adaptor molecule interferon promoter stimulator 1 (IPS-1) is another critical signalling protein regulating CHIKV-induced IRF3 activation and subsequent transcription of IRF3-dependent antiviral genes, including type I IFN [47]. IRF3 and IRF7 are key transcription factors involved in the induction of IFN-α/β that act downstream of RIG-I/MDA5/IPS-1, TLR3/TRIF and TLR7/MyD88 IFN induction pathways. Infection of a single (IRF3^−/−^ and IRF7^−/−^) and double (IRF3/7^−/−^) knockout mice clearly demonstrates that either IRF3 or IRF7 is required for survival following CHIKV infection, with IRF3/7^−/−^ mice showing significant viral dissemination with high titres [39]. Results suggest that IRF7 is the main transcription factor involved in IFN-α/β production after CHIKV infection [39]. Additionally, IRF7^−/−^ mice infected with SINV died 5–8 days after infection due to immune-mediated neurotoxicity associated with failure to regulate the production of inflammatory cytokines [48]. SINV infected IRF3^−/−^ mice developed persistent neurological defects and spinal cord inflammation but survived infection [48]. A number of studies that examined the relative contribution of IFN induction pathways demonstrate that RIG-I and MDA5 signalling via IPS-1 is the most important for IFN-α/β production in response to CHIKV infection, followed by the TLR3/TRIF pathway and the MyD88-dependent pathway [22,39]. The IPS-1-dependent sensing pathway was also shown to be the predominant regulator of antiviral activity induced by neurovirulent SINV [37].

A number of studies have outlined the sensitivity of the alphavirus-induced IFN response to fluctuations in temperature, with reduced antiviral efficacy at subnormal cellular temperatures and increased efficacy at supranormal temperatures [49,50]. Temperature-dependent transcription of IFN response genes leads to altered IFN-α/β and IFN stimulated gene (ISG) protein levels. RNA-Seq data suggest that increasing the housing temperature of mice from 22 °C to 30 °C increased the activation of the unfolded protein response (UPR) and antiviral type I IFN responses [50]. CHIKV replication and foot arthropathy were dramatically reduced in CHIKV-infected mice housed at 30 °C rather than 22 °C [50]. Conversely, a separate study demonstrated that lowering the temperature, at which mice were housed, exacerbated replication and disease in CHIKV-infected mice [49]. Both effects were dependent on type I IFN responses. These observations may explain why the symptoms of the arthritogenic alphaviral disease are largely reported in the joints of the extremities and raise the potential to use hyperthermia as a therapeutic.

### 2.2. Anti-Alphavirus Interferon Response Pathways

Following alphavirus detection and subsequent induction of the IFN pathway, IFN binds to cell surface receptors initiating the canonical JAK-STAT signalling cascade that leads to the transcriptional regulation of IFN-stimulated genes (ISGs). Large scale overexpression screening, RNAseq and microarray analysis has identified numerous ISGs with antiviral activity specific to alphaviruses [51,52,53,54,55,56]. Using the prototype arthritogenic alphavirus SINV as a model of IFN induction, antiviral activity against infection was shown to be only partially dependent on the combined contribution of the two major dsRNA-triggered antiviral pathways, the coupled 2’,5’-oligoadenylate (2-5A) synthetase (OAS)/RNase L pathway and dsRNA-dependent protein kinase PKR pathway [55]. OAS3 has a predominant role in RNase L activation and inhibition of SFV, SINV and CHIKV replication [57,58]. IFN signalling was sufficient to protect PKR^−/−^ or PKR/RNase L^−/−^ mice from lethal infection by up-regulating the expression of other antiviral proteins. Furthermore, blockade of global cellular translation during alphavirus infection has been shown to be independent of virus-induced phosphorylation of eukaryotic initiation factor subunit 2α (eIF2α) by PKR [47,59]. These results suggest the need for a level of redundancy in the activities of the IFN system to inhibit virus replication at multiple levels.

Multiple gene candidates for PKR/RNase L-independent IFN-induced anti-alphaviral activity have been identified [59]. Further analysis of these candidates in vitro highlighted ISG20 and zinc finger antiviral protein (ZAP) as potent inhibitors of SINV replication. ISG20 inhibits translation of CHIKV genomic RNA, and translation inhibition was associated with ISG20-induced up-regulation of over 100 other known antiviral effectors [60]. ZAP also effectively inhibits the replication of SINV, SFV and RRV by blocking the translation of incoming viral RNA [61]. Multiple ISG’s have been shown to have a synergistic effect with ZAP to enhance its antiviral activity [62,63]. TRIM25, an E3 ubiquitin and ISG15 ligase, is critical for ZAP’s ability to inhibit translation of the incoming SINV genome [62].

For positive-sense RNA viruses, like arthritogenic alphaviruses, translation is the first step required for viral replication. These viruses are therefore more likely to be sensitive to the effects of a block on translation than to PKR or RNase L, which are activated by dsRNA regions within viral RNA replication intermediates [59]. PKR-independent IFN-induced translation inhibition was shown to act on m(7)G cap-dependent translation at a step after association of cap-binding factors and before the formation of the 80S ribosome [64]. Interestingly, translation inhibition targets mRNAs that cross the cytoplasmic membrane, such as the SINV genome, leaving nuclear-transcribed RNAs unaffected and thus maintaining the expression of antiviral and stress-responsive genes [64].

Further validation of the anti-alphavirus activity of candidate ISGs was performed by overexpression of the protein using alphavirus vectors [54,65,66]. Viperin, p56 and ISG15 exhibited modest replication inhibition against SINV in vitro. The long isoform of PARP12 (PARP12L) demonstrated an inhibitory effect on the replication of SINV and CHIKV [65]. Viperin and ZAP profoundly attenuated SINV virulence in mice [54]. Viperin was highly induced in blood monocytes of CHIKV-infected patients. Furthermore, mice lacking viperin had higher viraemia and severe joint inflammation upon CHIKV infection compared to WT mice [67]. Anti-CHIKV functions of viperin were dependent on its localisation to the endoplasmic reticulum, and viperin expression in both hematopoietic and non-hematopoietic cells was instrumental in reducing CHIKV disease severity associated with CD4^+^ T cells [67,68]. The ISG transcription factor, promyelocytic leukaemia zinc finger protein (PLZF), has been shown to induce key antiviral mediators, including OAS1 and viperin [69]. Consequently, PLZF-deficient mice were more susceptible to SFV infection [69].

ISG15^−/−^ mice exhibited increased susceptibility to SINV and CHIKV infection [19,70]. ISG15 decreased SINV replication in multiple organs without inhibiting the spread of the virus throughout the host [66]. No differences in viral loads were observed between WT and ISG15^−/−^ mice infected with CHIKV; however, a dramatic increase in proinflammatory cytokines and chemokines was observed in ISG15^−/−^ mice [19]. These results suggest that ISG15 functions as an immunomodulatory molecule, and unchecked, the innate immune response can produce pathologic levels of immune mediators in response to alphavirus infection.

Additional ISG’s have been shown to contribute to the control of arthritogenic alphavirus infection. IFITM3 is a small transmembrane ISG shown to restrict pH-dependent viral fusion [71]. CHIKV, SFV, SINV and ONNV replication was enhanced in mouse fibroblasts lacking IFITM3. CHIKV-infected IFITM3^−/−^ mice sustained greater joint swelling than WT infected mice, and increased disease severity correlated with higher levels of proinflammatory cytokines and viral burden [71]. Tetherin/BST2 is an ISG and membrane protein that efficiently inhibits the release of SFV and CHIKV particles from host cells [72]. In vivo, BST-2 deficiency increased viraemia in CHIKV-infected mice and suppressed the innate response suggesting that BST2 regulates the CHIKV-induced inflammatory response [73].

## 3. Proinflammatory Cellular Responses During Acute Alphavirus-Induced Arthritic Disease

Infection with arthritogenic alphaviruses induces a sequence of cellular immune responses which together contribute to viral clearance. Alphaviruses are transmitted into the skin of a vertebrate host from an infected mosquito and are believed to infect skin-resident antigen-presenting cells (APCs) such as dendritic cells (DCs) or disseminate passively through the circulation [74]. While comprehensive studies elucidating the precise mechanism of alphaviral dissemination from the skin are lacking, it is hypothesised that alphaviruses disseminate through the skin via infected DCs or other skin-resident APCs (e.g., Langerhans cells), before migrating to skin-draining lymph nodes [75,76]. From the lymph nodes, alphaviruses migrate towards target musculoskeletal tissues, such as muscle, joints and synovial tissue [74]. A potent inflammatory response is then initiated in these target tissues, resulting in a large influx of infiltrating cells and the potent induction of antiviral and proinflammatory cytokines and chemokines. The major component of this cellular infiltrate are myeloid cells, such as inflammatory monocytes and macrophages [77,78,79], but also consists of neutrophils [80], T lymphocytes and natural killer (NK) cells [74,81]. These inflammatory cellular infiltrates occur in the muscle and joints in RRV-infected [82] and CHIKV-infected mice [81] and in blood and synovial fluid of RRV-infected [79] and CHIKV-infected patients [78]. However, the phenotype and function of these cell subsets in alphavirus infection remains poorly understood. Arthritogenic alphaviruses have been shown to infect cells in skeletal muscle and synovial tissue, which, in turn, contributes to an inflammatory response leading to cellular hyperplasia and tissue damage. In addition, this inflammatory response indirectly up-regulates proinflammatory cytokines, thus exacerbating inflammation through an inflammatory cascade [82,83].

In addition to a prominent antiviral type I IFN immune response, the acute stage of arthritogenic alphavirus infection is associated with a significant release of inflammatory cytokines in target tissues. Plasma from acute-stage (<21 days post-infection) CHIKV patients, revealed significantly elevated levels of proinflammatory cytokines (IL-1, IL-6, IL-8 and IL-16) and chemokines (CCL2, CCL4 and CXCL10), as well as regulatory cytokines (IL-1RA, IL-10 and IL-13) and growth factors (G-CSF, GM-CSF, VEGF and SCGF) [14,78,84,85,86,87,88]. Furthermore, potent proinflammatory cytokines tumour necrosis factor (TNF)-α and IFN-γ, as well as CCL2, IL-4, IL-7, IL-8, CXCL10 and GM-CSF, are also up-regulated in RRV-infected cells in vitro [84] and in the serum of RRV-infected patients, compared to healthy controls [89]. IL-6, TNF-α and IFN-γ levels are also elevated in the synovial fluid of patients infected with RRV [90], as well as in patients infected with CHIKV [91] and MAYV [92]. This indicates a common immunopathological proinflammatory cytokine profile shared among arthritogenic alphaviruses.

This influx of cytokines increases the recruitment of macrophages and monocytes [78,81,82] and triggers antiviral responses from CD8^+^ T cells and NK cells [14,93]. This inflammatory cascade and resulting local tissue inflammation and damage are associated with the development of the arthritic disease. Interestingly, a recent 2018 study revealed an early cytokine response in CHIKV patients (TNF-α, IL-2, IL-4 and IL-13) correlated with a reduction in chronic arthritic disease development [94]. Subsequently, a weak cytokine response during the acute phase was predictive of the development of chronic disease. This suggests that a strong early cytokine response in CHIKV infection may aid in viral clearance.

### 3.1. Macrophage-Derived Immune Responses

Monocytes and macrophages comprise a major component of the cellular infiltrate in alphavirus-infected tissues [77,78,79]. CHIKV has been shown to infect monocyte/macrophages, which subsequently infiltrate the lymph nodes [78], as well as infect synovial macrophages [77]. Macrophages and macrophage-derived proinflammatory cytokines play an important role in inducing pathology in both muscle and synovial tissues. Namely, macrophage-derived TNF-α and CCL2 are elevated in the muscle and joints of RRV-infected mice [90]. Further, experimental depletion of macrophages in RRV-infected mice was associated with reduced tissue damage compared with untreated infected mice [90]. This suggests that macrophages and macrophage-derived products play a key role in RRV-induced arthritic and myositic disease. In addition to the acute cellular infiltrate, macrophages are generally viewed as the main reservoirs during chronic CHIKV infection and thus have the potential to sustain viral persistence in the host [77]. 

CCR2 (the receptor for CCL2) is suggested to play a major role in the recruitment of monocytes and macrophage in alphavirus infections. Namely, CHIKV-infected mice deficient in CCR2 demonstrated exacerbated and prolonged inflammation and bone damage [80]. Interestingly, in CCR2^−/−^ mice, the monocyte/macrophage dominant infiltrate switched to a severe neutrophil (followed by an eosinophil) infiltrate. This switch was associated with increases in several proinflammatory mediators (IL-1β, CXCL1, CXCL2, G-CSF) and IL-10 in CCR2^−/−^ mice compared with WT mice [80]. These findings may contrast with earlier studies which showed that the inhibition of CCL2 using Bindarit (a pan-CCL inhibitor) was associated with reduced arthritic and myositic inflammation in RRV-infected and CHIKV-infected mice [95,96]. Therefore, while CCL2-producing monocytes/macrophages seem to contribute to alphavirus-induced inflammation and may be an appropriate target for therapeutic intervention, disruption of CCR2 signalling may, in turn, enable alternative neutrophil-driven mechanisms that are detrimental to the host. Interestingly, a recent study by Diamond and colleagues identified monocyte Fc receptors as an essential component of anti-CHIKV monoclonal antibody (mAb)-dependent neutralisation leading to a reduction in CHIKV-induced inflammation in an experimental mouse model of CHIKV disease [97]. This indicates that while monocytes are important in promoting antiviral responses and acute inflammation, they are required for effective mAb-mediated clearance and clinical protection in CHIKV disease. Thus, further studies focusing on the role of monocytes and the CCR2-CCL2 axis in arthritogenic alphavirus infection would help design better targeting strategies to limit monocyte/macrophage-driven musculoskeletal tissue inflammation and disease.

### 3.2. T-Cell-Derived Immune Responses

Although myeloid cells, such as inflammatory monocytes and macrophages, are a major component of inflammatory infiltrates associated with alphavirus infection, the adaptive immune response has been shown to play an important role in tissue inflammation. Adaptive immunity, which comprises of T and B lymphocytes and NK cells, can have both a protective effect (antiviral responses) and detrimental consequences (exacerbated inflammation) for the infected host [74]. Although studies have shown that inflammation induced by alphavirus infections is largely driven by innate immune responses via type I IFN signalling [22,39], these alone are insufficient in completely eradicating virus, as CHIKV antigen has been shown to persist in tissues of CHIKV-infected mice after IFN-α/β levels had returned to baseline levels and viraemia had subsided [77,98]. Antibody-mediated depletion of NK cells alone has been shown to have no effect on viral loads or disease severity in RRV-infected mice [99,100]. CD4^+^ and CD8^+^ T cells have been identified as key components of the cellular infiltrate in musculoskeletal tissue in RRV and CHIKV-infected mice [81,82,101,102,103], as well as RRV and CHIKV-infected patient serum and synovial and muscle tissue [104,105,106]. It has been suggested that T cells may not be directly implicated in viral control, as CHIKV viraemia was identical in RAG2^−/−^ (mice lacking functional B and T lymphocytes), CD4^−/−^ and CD8^−/−^ compared to WT mice [102]. This is consistent with the findings demonstrating clearance of CHIKV viraemia at day 4 post-infection observed before peak joint swelling in the mouse model of CHIKV disease at day 7 [81,101]. This also suggests that CHIKV-induced inflammation and arthropathy are more dependent on host immune responses, rather than viral persistence. 

Specifically, distinct CD4^+^ T cell subsets have been identified in alphavirus-induced inflammation. Th1-associated genes encoding IFN-α, IFN-γ and TBX21, Th2 genes encoding IL-4, IL-5 and IL-6 and Th17-associated gene encoding IL-17A are significantly expressed in CHIKV infection [107]. This suggests that Th1, Th2 and Th17 cells and their secreted cytokines play a role during CHIKV infection. In contrast, MAYV infection is associated with high levels of regulatory T cells (Tregs), a subset of CD4^+^ T cells that display regulatory/suppressive properties during immune responses and inflammation [107]. In CHIKV-infected mice, Tregs have been shown to interact with antigen-presenting DCs, which induced down-regulation of DC-derived costimulatory signalling. This, in turn, limited the expansion of CHIKV-specific CD4^+^ T cells and thereby reduced CHIKV-induced joint swelling [108]. Therefore, T cell subsets may play different roles in alphavirus infection, with some producing proinflammatory cytokines and others lending a more protective role.

## 4. Immune Responses During Chronic Alphavirus-Induced Arthritic Disease

In the acute phase of alphavirus infection, immune responses initiate tissue inflammation and pathology in the muscles and synovial tissue which usually resolve over time, and after which patients generally recover. However, in some patients, the alphaviral disease can develop into a chronic stage (>3 months after onset of infection) with persistent inflammation and chronic arthritic disease. The mechanisms which contribute to the transformation of acute disease into chronic disease are still unclear, though it has been hypothesised that chronic inflammation could arise from the persistence of viral antigen in tissues, or residual proinflammatory mediators that remain in previously infected tissues and may continue to trigger inflammatory responses [74]. The live virus has been shown to persist in the joint tissue of CHIKV-infected μMT mice lacking mature B cells, suggesting that virus-specific antibody is required for clearance of infection [109].

Serum from convalescent (recovering) CHIKV patients with symptomatic arthralgia (>30 days post-disease onset) showed elevated levels of IL-1β, IL-5, IL-10, IL-12, IFN-γ and TNF-α compared to healthy controls [86]. Interestingly, some cytokines increase or decrease during the progression from the acute phase to the convalescent phase. During acute disease, IL-1β, IL-2, IL-5, IL-12 and TNF-α were initially low but increased as patients recovered from the disease. In contrast, IL-6, CXCL9, CXCL10, CCL2 were elevated in the acute phase but decreased as patients recovered [86]. 

Several studies have shown that IL-6 is elevated in patients with chronic CHIKV disease [85,87,104]. In addition, IL-6 levels can be greater in the chronic phase compared to the acute phase [87], and during chronic CHIKV disease, reside in greater concentrations in the inflamed joint compared to patient plasma [104]. It is suggested that IL-6 contributes to alphavirus-induced bone loss by stimulating the release of receptor activator of nuclear factor kappa B (RANKL) [110] while inhibiting its soluble decoy receptor osteoprotegerin (OPG), thus promoting osteoclastogenesis, a bone resorption mechanism driven by osteoclasts and bone loss [111,112]. Furthermore, inhibition of IL-6 in RRV-infected mice was shown to ameliorate bone loss, indicating that IL-6 could be strongly associated with chronic alphavirus-induced arthritic disease and associated bone pathology [113]. 

Other cytokines associated with the chronic phase of alphavirus-induced disease are IL-8, IL-12, GM-CSF, CCL2, CCL3 and CCL4 [85,87]. IL-12, a classical Type 1 proinflammatory and regulatory cytokine generally involved in acute inflammatory responses, remained elevated in chronic CHIKV disease patients but reverted to baseline levels in patients who had recovered [104]. Of note, IL-17, a cytokine that has been well described in RA and which is known to exacerbate inflammation and bone loss, only became detectable in the chronic phase in CHIKV patients [85]. It is, therefore, possible that the prolonged release of these cytokines and proinflammatory mediators could be one of the underlying mechanisms for the development of the chronic alphavirus-induced disease. In addition, continued viral replication and persistence in synovial tissue several weeks post-disease onset could aid in maintaining a prolonged inflammatory response in the chronic phase of disease [114].

## 5. Treatment Strategies Targeting Alphavirus-Induced Inflammation

There are currently no licensed treatments or vaccines that are specific for alphavirus infection. The majority of patients are prescribed analgesics and/or non-steroidal anti-inflammatory drugs (NSAIDs) [115]. However, adverse immunomodulatory effects have been associated with NSAID use. Namely, NSAIDs have been demonstrated to inhibit B lymphocytes (a source of neutralising IgM and IgG antibodies), which can consequently suppress antibody production and thereby may impair host immune defence and long-term antiviral immunity [116]. Therefore, the development of specific treatments targeting alphaviruses remains of vital importance, particularly, in chronic disease patients. Among promising therapeutic candidates is Bindarit, an inhibitor of monocyte/macrophage-derived CCL2, CCL7 and CCL8 (Figure 1). While treatment with Bindarit resulted in ameliorated arthritic disease and bone loss in RRV-infected mice, human clinical trials are required to ascertain the efficacy of such targeted approaches [95,96,117].

Pentosan polysulfate (PPS) has been shown to maintain cartilage proteoglycan levels in non-infectious arthritis models and is currently approved in the US for interstitial cystitis in humans and osteoarthritis in horses and dogs [118,119]. PPS treatment in RRV-infected mice reduced inflammatory infiltrates and joint pathology [120]. Similarly, PPS treatment in CHIKV-infected mice reduced IL-6, IL-9, G-CSF and CCL2 and decreased joint swelling [120]. While the exact mechanism by which PPS modulated joint inflammation in alphavirus-infected mice is unknown, it is suggested that its ability to reduce inflammation is due to its inhibitory effect on IL-6 and CCL2. Furthermore, PPS treatment resulted in an increase in IL-10 levels, thereby promoting anti-inflammatory responses [120]. PPS is currently in phase II clinical trials for patients diagnosed with RRV-induced arthritic disease. Another treatment strategy that has been investigated is Etanercept (Enbrel), a fusion human Fc-TNF antibody currently used to treat patients with RA and other forms of non-infectious arthritis [121]. However, Etanercept treatment in RRV-infected mice exacerbated disease and tissue damage, increased inflammatory cell infiltrate and elevated viral titres [122]. Therefore, Etanercept is not recommended as a potential treatment strategy in alphavirus-induced disease. Taken together, this suggests that potential treatment candidates that target proinflammatory mediators that are released in alphavirus infections could help prevent virus-induced inflammation and arthritic disease. The disease-modifying anti-rheumatic drug Methotrexate, routinely used to treat patients with RA, was shown to exacerbate acute disease in RRV-infected mice, increasing inflammatory cell infiltrates and virus titres in skeletal muscle [123]. Interestingly, Methotrexate may have found utility in treating chronic inflammatory rheumatism post-CHIKV infection [124].

In addition to targeting proinflammatory mediators, drug treatments targeting T cell responses have also shown promise in reducing alphavirus-induced disease. Recently, treatment with Fingolimod (FTY720), a T cell suppressor which blocks T cell egress from lymphoid organs to peripheral tissues, reduced T cell accumulation in the joints of CHIKV-infected mice, thereby reducing joint inflammation and disease [125]. Another potential drug treatment is Abatacept (CTLA4-Ig), which inhibits T cell activation by blocking APC secretion of costimulatory signals to T cells. In CHIKV-infected mice, Abatacept, in combination with a neutralising anti-CHIKV monoclonal antibody, effectively reduced T cell accumulation in joints, ameliorated joint inflammation and decreased proinflammatory mediators, including CXCL10, CCL2, CCL4 and CCL5. In contrast, monotherapy with Abatacept resulted in reduced levels of CXCL10 and CCL4, whereas treatment with anti-CHIKV antibody alone decreased CCL2 only [126]. This suggests that combination therapy of antibody-derived antiviral drugs and proinflammatory antagonists may serve as a potential treatment strategy in patients with the alphavirus-induced arthritic disease.

## 6. Concluding Remarks

The spread of arthritogenic alphaviruses and their mosquito vectors represents a major public health problem globally with severe social and economic impacts. For example, 106,592 cases were reported following the 2014 CHIKV outbreak in Colombia with the estimated total disability-adjusted life years (DALYs) of 40.44 to 45.14 lost/100,000 population [127]. The outbreak was estimated to cost at least US$73.6 million [127]. With no targeted therapeutics or vaccines to prevent arthritogenic alphaviral disease, substantial effort has been made to decipher the mechanisms of alphavirus-induced inflammation and disease. Much of this research has highlighted the protective and pathogenic role of the host immune system during infection. As one of the first responses to alphavirus infection, IFN induction is stimulated by a varied network of PRRs and downstream activators. However, the predominant regulator of antiviral activity and the relative contribution of IFN signalling pathways during alphavirus infection remains unclear. Large scale screens of ISGs stimulated during alphavirus infection provide a useful starting point to identify antiviral effectors, but more in-depth analysis of potential candidates is required. Analysis of animal models of alphavirus infection supplemented with clinical samples has provided a detailed evaluation of the immunopathology associated with arthritogenic alphavirus infection. This has led to the identification of novel therapeutic targets to specifically ameliorate alphavirus-induced inflammation. A number of candidate drugs that exploit these targets have shown promising results in preclinical trials, with some progressing to phase II human trials. The chronic debilitating impact of alphaviral disease creates a high economic burden, yet the mechanisms responsible for the chronic rheumatic manifestations of alphaviral disease remain unresolved. It is, thus, imperative to fully elucidate the immune effectors responsible for controlling alphavirus infection and disease.

## Figures and Tables

**Figure 1 viruses-11-00290-f001:**
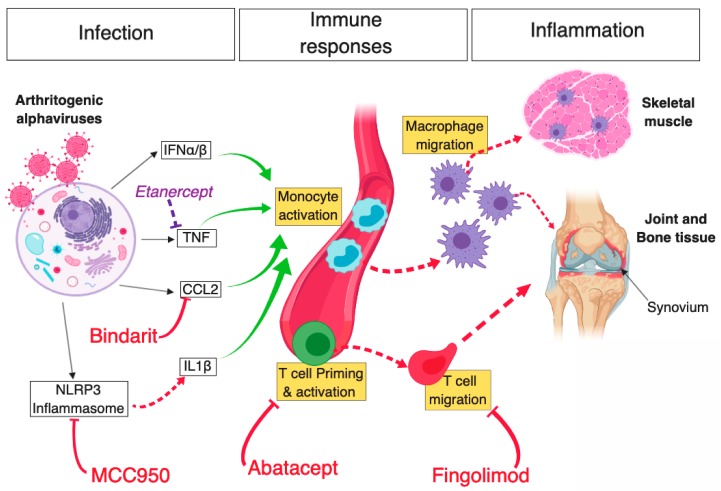
Schematic diagram of key mechanisms of arthritogenic alphavirus-mediated musculoskeletal pathology and novel therapeutic strategies that were shown to limit inflammation and disease. Infection of host mammalian cells by arthritogenic alphaviruses—following a mosquito bite—elicits a potent antiviral cellular response driven by type I IFN, subsequently leading to the production of various proinflammatory cytokines (e.g., CCL2, TNF). In parallel, activation of the NLRP3 inflammasome following alphaviral infection leads to the production of IL-1β, which further contributes to inflammation. Bindarit, a monocyte-chemotactic protein (MPC) inhibitor with CCL2-neutralising properties, was shown to curtail monocyte activation and subsequent macrophage migration to musculoskeletal tissues. Likewise, inhibition of NLRP3 using MCC950 was found to abrogate inflammasome and caspase-driven IL-1β production, in turn, reducing muscle inflammation and bone resorption in CHIKV-infected mice. However, inhibition of TNF using Etanercept (commonly used in the treatment of rheumatoid arthritis) exacerbated alphaviral inflammation of muscle and joint tissue. In addition, alphavirus infection leads to the activation of adaptive immune responses and inhibition of CD4^+^ T cell priming, or egress from draining lymph nodes in CHIKV-infected mice using Abatacept and Fingolimod, respectively, led to a substantial reduction in cellular infiltration in ankle joints. Red T arrows: inhibitory effect leading to positive disease outcome; Dashed red arrows: direct activation; Solid green arrows: cell stimulatory effect; Solid black arrows: antiviral response products; Purple T arrow: inhibitory effect leading to negative disease outcome. Diagram created using BioRender©.

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
