# Peer review of "Arthritogenic Alphavirus-Induced Immunopathology and Targeting Host Inflammation as A Therapeutic Strategy for Alphaviral Disease"

_viruses, 2019, doi:10.3390/v11030290_

Round 1

Reviewer 1 Report

Mostafavi et al have contributed a complete and useful review of the immunopathology induced by arthritogenic alphavirus with focus on innate immunity to virus infection and inflammation pathway. They initiate their paper by explaining the difference between arthritogenic and modern alphaviruses and the clinical symptoms caused by arthritogenic alphavirus infection. This is a good example for those not familiar with alphavirus. They then provide a detailed innate immune system activated by alphavirus infection. This part of the paper provides examples of different mechanisms that host cells use to limit the viral infection. The third part of the paper talks about the immune responses transformed from the acute phase to chronic infection. At the end the authors point out that combination therapy of antibody and proinflammatory mediators may be the potential key for the future treatment.

Overall the manuscript is well written and clear. There is only a minor point:

Line 12:  explosive outbreaks with some viruses - a major global concern.

Author Response

We thank the reviewer for their constructive comments and input into the manuscript “Arthritogenic alphavirus-induced immunopathology and targeting host inflammation as a therapeutic strategy for alphaviral disease”. We have addressed all reviewer concerns below.

Line 12:  explosive outbreaks with some viruses - a major global concern.

Response – Grammar in Line 12 has been edited and now reads “explosive outbreaks, with some viruses a major global concern”.

Reviewer 2 Report

It is a pleasure to read the manuscript “Arthritogenic alphavirus-induced immunopathology and targeting host inflammation as a therapeutic strategy for alphaviral disease” submitted by Helen Mostafavi et al to Viruses. This is a neatly written review describing the Old World arthrogenic alphavirus-induced immunopathology. The review also explored some of the therapeutic strategy against alphaviral disease targeting the host inflammation.

Major:

1.      The NK cell response and B cell/antibody response should be expanded.  

2.      It would be necessary to expand the research of the immunopathology related to alpha virus infection in humans.

Author Response

We thank the reviewer for their constructive comments and input into the manuscript “Arthritogenic alphavirus-induced immunopathology and targeting host inflammation as a therapeutic strategy for alphaviral disease”. We have addressed all reviewer concerns below.

1.    The NK cell response and B cell/antibody response should be expanded.  

The following passages and references have been added to the manuscript:

Line 350: “Antibody-mediated depletion of NK cells alone has been shown to have no effect on viral loads or disease severity in RRV-infected mice [99,100].” 

99. Haist, K.C.; Burrack, K.S.; Davenport, B.J.; Morrison, T.E. Inflammatory monocytes mediate control of acute alphavirus infection in mice. PLoS pathogens 2017, 13.

100. Burrack, K.A.S.; Hawman, D.W.; Jupille, H.J.; Oko, L.; Minor, M.; Shives, K.D.; Gunn, B.M.; Long, K.M.; Morrison, T.E. Attenuating Mutations in nsP1 Reveal Tissue-Specific Mechanisms for Control of Ross River Virus Infection. Journal of virology 2014, 88, 3719-3732.

Line 381: "Live virus has been shown to persist in the joint tissue of CHIKV-infected μMT mice lacking mature B cells, suggesting that virus-specific antibody is required for clearance of infection [109]."

109. Hawman, D.W.; Fox, J.M.; Ashbrook, A.W.; May, N.A.; Schroeder, K.M.S.; Torres, R.M.; Crowe, J.E.; Dermody, T.S.; Diamond, M.S.; Morrison, T.E. Pathogenic Chikungunya Virus Evades B Cell Responses to Establish Persistence. Cell Rep 2016, 16, 1326-1338, doi:10.1016/j.celrep.2016.06.076. 

2.    It would be necessary to expand the research of the immunopathology related to alpha virus infection in humans.

The following passages and references have been added to the manuscript:

Line 73: Specifically, IFN-α levels were significantly elevated in plasma of acute CHIKV-infected patients [Wauquieret al.].

Wauquier, N.; Becquart, P.; Nkoghe, D.; Padilla, C.; Ndjoyi-Mbiguino, A.; Leroy, E.M. The Acute Phase of Chikungunya Virus Infection in Humans Is Associated With Strong Innate Immunity and T CD8 Cell Activation. The Journal of Infectious Diseases 2011, 204, 115-123, doi:10.1093/infdis/jiq006.

Line 281:These inflammatory cellular infiltrates occur in the muscle and joints in RRV-infected [82] and CHIKV-infected mice [81] and in blood and synovial fluid of RRV-infected [79] and CHIKV-infected patients [78].

Line 304:Interestingly, a recent 2018 study revealed an early cytokine response in CHIKV patients (TNF-α, IL-2, IL-4 and IL-13) correlated with a reduction in chronic arthritic disease development.

Line 395: Several studies have shown IL-6 is elevated in patients with chronic CHIKV disease.

Reviewer 3 Report

This is an extremely well written and comprehensive review of the innate immune response to acute alphavirus infection and alphavirus-induced immunopathology. Immune responses during chronic alphavirus-induced arthritic disease, as well as treatment strategies targeting alphavirus-induced inflammation are also discussed. Job well done.

One minor typo was detected; line 164- change "associated" to "associate."

Author Response

We thank the reviewer for their constructive comments and input into the manuscript “Arthritogenic alphavirus-induced immunopathology and targeting host inflammation as a therapeutic strategy for alphaviral disease”. We have addressed all reviewer concerns below.

One minor typo was detected; line 164- change "associated" to "associate."

Response – In line 164 "associated" has been changed to "associate."

Reviewer 4 Report

In this manuscript, Mostafavi et al . reviewed the recent findings of immune responses to alphaviruses with the ultimate goal of exploring the possibility of targeting host responses to develop therapeutics. The authors tried very hard to cover a lot of important topics related to inflammatory responses elicited by alphavirus infections. It is a well-written manuscript. What is really missing is a coherent story and some transition phrases to connect different topics.

Major concern:

Although the authors lay out the focus of this review on innate immune responses and the application of current knowledge for the development of treatments, I have some concern over the fact the authors left out the importance of adaptive immune responses such humoral immunity in limiting alphavirus infections. For instance, there has been a lot of work related to neutralizing monoclonal antibodies against chikungunya virus (CHIKV). Therapeutic antibodies can indeed be the forerunners in the preclinical development of specific treatments for CHIKV (see L439-41).

If the authors decided to focus on innate immune responses (see L17), why did the authors also include the discussion of T-cell immune responses? This made the contents after L258 rather confusing. Perhaps the authors need to update the abstract.

L12-3 and 397: The authors should acknowledge the fact that there are veterinary vaccines for equine encephalitis viruses in the United States available if they decided to write broadly about alphaviral diseases.

L34: Mayaro virus (MAYV) is an alphavirus endemic in the Latin America (New World). Describing it as an Old World alphavirus is somewhat strange.

L42-4: Transmission of CHIKV in the Americas mainly involved Aedes aegypti. Describing the importance of Ae. albopictus right after the description of outbreaks in America and using the term "This epidemic" can be misleading. The authors also provided the citation related to outbreaks in the Indian Ocean basin, which is part of the Old World.

The authors provided quite a few references about Sindbis virus (SINV). Although these are peer-reviewed studies, I would encourage the authors to reconsider the conclusions made from studies using neuroadapted variants of SINV.

Minor editorial changes

L12: Update "some viruses a major global concern." This phrase doesn't quite make sense.

L26: What is the purpose of specifying "non-human"? Didn't the authors just try to describe the importance of reservoirs in maintenance cycles such as enzootic and sylvatic transmissions?

L38: Global is a strange word here. Perhaps emerging and re-emerging will be a better adjective.

L41: "CHIKV recently become......" does not make sense.

L43: Change all A. to Ae.  in this manuscript. Normally, Aedes is abbreviated as Ae. Although the authors can argue this makes no difference in an immunology paper, the authors must remember that O'nyong-nyong virus, another Old World alphavirus, is transmitted by Anopheles species mosquitoes. This abbreviation can be confusing.

L61-2: I know the authors want to talk about Old World Alphavirus but cannot agree with this sentence as it is written. Alphavirus infections with neurological diseases can be quite life-threatening if it involves specific types of Eastern equine encephalitis virus.

L71-3: The authors cited several review articles in this part of the manuscript. I recommend just including the research articles, which collectively led to the conclusion that type-I interferon (IFN) is important for the host immune responses against CHIKV.

L81: A phrase that helps the transition of the topic from type-I IFN to complements and other effectors.

L103: It will be helpful to review the roles of other interferon regulatory factors (IRFs) in addition to IRF-1. There have been several IRF knockout models used as CHIKV pathogenesis models.

L110: Provide the full name of PAMP.

L111-3: The sentence "During......host PRRs." does not make sense.

L111: It should be "single-stranded" RNA.

L113: Provide the full name of PRR.

L127: There should be a comma after "Again."

L131: There should be a comma after "Interestingly."

L149: It should be "Myd88-deficient" mice.

L169: Viremia can only be induced in vivo. The authors need to pick one word.

L180-3: This is a long sentence. I really cannot understand what the authors are trying to describe.

L196: The authors might need to be careful about suggesting manipulating the body temperature of vertebrate hosts as a therapeutic strategy. I do not see any citations to support this argument.

L220 and 242-8: Why did the authors mention ISG15 and ISG15 ligase in two different paragraphs? Is there any specific reason not to combine them in one paragraph? The authors should consider including the role of ISG15 in limiting CHIKV infection. The finding was published prior to the SINV study cited by the authors.

L259-78: This is a rather basic description of the dynamics of infection. It will certainly be more appropriate to move it to the beginning of section 2, which began with a rather deep diver into type-I IFN responses.

L328: Provide the citation for the monoclonal antibody (mAb)-mediated clearance. And, also provide the full name of mAb.

L350: Can the authors provide some comments in the pathogenic roles of host immune responses and viral persistence? The paragraph earlier suggested the persistent infection of macrophages might also be important and is contradictory to what is written here.

L351-61: It is hard to understand this paragraph. Are the authors trying to show the importance of regulatory T cells?

Author Response

L41: "CHIKV recently become......" does not make sense.

L41 “become” has been changed to “became”

L43: Change all A. to Ae.  in this manuscript. Normally, Aedes is abbreviated as Ae. Although the authors can argue this makes no difference in an immunology paper, the authors must remember that O'nyong-nyong virus, another Old World alphavirus, is transmitted by Anopheles species mosquitoes. This abbreviation can be confusing.

L43 “A.” has been changed to “Ae.

L111-3: The sentence "During......host PRRs." does not make sense.

L43: Grammatical changes have resolved this issue.

L111: It should be "single-stranded" RNA.

L43: “single strand” has been changed to “single-stranded”.

L127: There should be a comma after "Again."

L127: a comma has been added

L131: There should be a comma after "Interestingly."

L131: a comma has been added

L149: It should be "Myd88-deficient" mice.

L149: “Myd88 deficient” has been changed to “Myd88-deficient” 

L169: Viremia can only be induced in vivo. The authors need to pick one word.

L169: All now read as “viraemia” and “in vivo” has been removed.